# Platelets, Not an Insignificant Player in Development of Allergic Asthma

**DOI:** 10.3390/cells10082038

**Published:** 2021-08-10

**Authors:** Liping Luo, Junyan Zhang, Jongdae Lee, Ailin Tao

**Affiliations:** The Second Affiliated Hospital, Guangdong Provincial Key Laboratory of Allergy & Clinical Immunology, State Key Laboratory of Respiratory Disease, Guangzhou Medical University, Guangzhou 510260, China; luoliping0503@163.com (L.L.); kuhn2000@163.com (J.Z.); j142lee@ucsd.edu (J.L.)

**Keywords:** allergic asthma, platelets, adaptive immune response, antiplatelet treatment, megakaryocytes

## Abstract

Allergic asthma is a chronic and heterogeneous pulmonary disease in which platelets can be activated in an IgE-mediated pathway and migrate to the airways via CCR3-dependent mechanism. Activated platelets secrete IL-33, Dkk-1, and 5-HT or overexpress CD40L on the cell surfaces to induce Type 2 immune response or interact with TSLP-stimulated myeloid DCs through the RANK-RANKL-dependent manner to tune the sensitization stage of allergic asthma. Additionally, platelets can mediate leukocyte infiltration into the lungs through P-selectin-mediated interaction with PSGL-1 and upregulate integrin expression in activated leukocytes. Platelets release myl9/12 protein to recruit CD4^+^CD69^+^ T cells to the inflammatory sites. Bronchoactive mediators, enzymes, and ROS released by platelets also contribute to the pathogenesis of allergic asthma. GM-CSF from platelets inhibits the eosinophil apoptosis, thus enhancing the chronic inflammatory response and tissue damage. Functional alterations in the mitochondria of platelets in allergic asthmatic lungs further confirm the role of platelets in the inflammation response. Given the extensive roles of platelets in allergic asthma, antiplatelet drugs have been tested in some allergic asthma patients. Therefore, elucidating the role of platelets in the pathogenesis of allergic asthma will provide us with new insights and lead to novel approaches in the treatment of this disease.

## 1. Introduction

Asthma is a chronic and heterogeneous pulmonary disease which affects over 300 million people around the world [1]. Asthmatic patients can vary widely in terms of clinical presentation, severity, and pathophysiology, though they generally experience similar symptoms such as coughing, wheezing, breathlessness, and reversible airway obstruction. While its prevalence used to be reserved mainly for high-income countries, it has become a major public health challenge in China due to economic advances as well as rapid changes in environment and lifestyle [2]. Due to the complexity of the disease, there are several classification standards. Asthma phenotypes can be causatively grouped into allergic asthma and nonallergic asthma, or cellularly into eosinophilic, neutrophilic, paucigranulocytic, and mixed granulocytic asthma [3]. Furthermore, it is divided into Type 2 asthma and non-Type 2 asthma based on the inflammatory cytokine profiles [4,5].

Platelets are anucleate blood cells with a diameter of 2~4 μm generated from megakaryocytes. They contain a variety of secretory granules, such as α-granules, dense granules (δ-granules), and lysosomes, which not only contain coagulation-related factors, but also inflammatory mediators and protease [6]. Platelets are well known for their roles in hemostasis and thrombosis, and now increasing evidence has highlighted their immunological roles in inflammation (Figure 1). Platelet malfunctions can lead to atherosclerosis, stroke, myocardial infarction, and deep venous thrombosis as well as allergic diseases, such as atopic dermatitis [7,8,9,10]. 

The involvement of platelets in the pathogenesis of asthma has been known for many years, especially in the development of allergic asthma and aspirin-exacerbated respiratory disease (AERD) [11,12]. Allergic asthma, also called atopic asthma, is the most common type of asthma with similar pathological features including mucus overproduction, chronic airway inflammation, airway remodeling, and airway hyperresponsiveness (AHR). It has been characterized as infiltration of eosinophils, mast cells, and lymphocytes in the airways and features increased secretion of type 2 cytokines such as IL-4, IL-5, and IL-13 after allergen exposure [13]. The onset of allergic asthma is influenced by both genetic backgrounds and environmental factors, and is closely related to allergic rhinitis [1,14]. Allergens that trigger allergic asthma include house dust mites, pollens, dander from cats and dogs, mold spores, cockroaches [15], and even small molecules such as toluene diisocyanate (TDI) [16].

In this review, to provide a comprehensive understanding of the relationship between platelets and allergic asthma, we will elaborate on the following aspects: (i) the current understanding of platelets’ involvement in allergic asthma; (ii) the underlying mechanism of platelets’ role in the initiation of adaptive immunity and the pathogenic development of allergic asthma; (iii) the antiplatelet therapy in asthma treatment; and (iv) new insights on platelets in asthma. 

## 2. Current Understanding of Platelets’ Involvement in Allergic Asthma

Platelet activation in allergic asthma has been highlighted by a great number of surveys (Table 1). Disturbed hemostatic balance in the lungs of allergic asthma has been evidenced by increased levels of cellular fibronectin, a marker of vascular injury in asthmatic patients [17,18]. It is said that the imbalances between the coagulation and anticoagulation system and the fibrinolytic system are jointly involved in asthma [18]. The platelet activation markers such as β-thromboglobulin (β-TG) and platelet factor 4 (PF4), which are increased in the plasma of atopic dermatitis patients, were found to be higher when patients were afflicted with concomitant asthma and allergic rhinitis [10,19]. By intrabronchially challenging house-dust-mite (HDM)-sensitive asthmatic patients with *Dermatophagoides pteronysisnus* (Dp) extract, it has been shown that allergen challenge is correlated with platelet activation in vivo, manifested as decreased levels of platelet count and increased plasma levels of β-TG and PF4 [20]. The involvement of platelets in allergic asthma can be indicated by the increased platelet counts in the bronchoalveolar lavage fluid (BALF) of the patients and animal models [21]. Consistently, the increased levels of PF4 and β-TG were also detected in BALF [22]. Furthermore, platelet deposition on interalveolar septum walls was also observed in bronchial biopsies and its number was significantly increased after allergen challenge [23,24]. Moreover, platelet-leukocyte conjugates were also observed in the peripheral blood of asthmatic patients after allergen exposure [25]. Platelet activity in patients with pollen-induced seasonal allergic rhinitis and asthma, using plasma PF4 as an indicator, was found to increase during the grass pollen season and decrease in the off season [26]. It has been shown that P-selectin in the nasal lavage fluid of asthmatic patients is positively correlated with the level of eosinophil cationic protein (ECP) [27]. Furthermore, eosinophil β1-integrin activation in asthma was found to be associated with activated platelets in a P-selectin-mediated manner [28,29]. Duarte, D. et al. found a significantly higher level of platelet-derived microparticles (PMPs) in the peripheral blood of patients with allergic asthma compared to healthy individuals [30].

The expression of high-affinity IgE receptor (FcεRI) and low-affinity IgE receptors (FcεRII/CD23) on platelets provides the structural basis for platelets’ involvement in allergic asthma. Monoclonal antibodies targeting IgE receptors (anti-FcεRI and anti-CD23) or IgE binding to the asthmatic patients’ platelets (anti-IgE) induced the RANTES release by platelets and cytotoxicity against schistosomula [31,34,35,37]. It has also been demonstrated that the allergen triggered platelet activation in an IgE-FcεR-dependent pathway and induced inflammatory mediators, such as serotonin (5-HT) and RANTES, from platelets [34,35]. Research conducted on humans revealed that the percentage of IgE^+^ platelets in atopic asthmatic patients was twice of that in humans with a normal IgE level, which is 10% [31]. This was also evidenced by the fact that the expression levels of FcεRI and FcεRII in platelets were significantly different between sham- and OVA-immunized mice [38]. However, others reported no distinct difference in the expression levels of FcεRI on the platelets in allergic patients and healthy controls [35]. 

In animal models, allergen exposure resulted in platelet migration to the airways, essential for eosinophil and lymphocyte recruitment and activation [25], and the critical role of P-selectin in platelets in the development of allergen-induced airway response was demonstrated in ovalbumin (OVA) and cockroach-induced murine asthmatic models [43,44]. Pitchford, S.C. et al. [38] showed that in OVA-sensitized mice, platelets migrated out of the blood vessels in an allergen-IgE-FcεRI pathway rather than due to hemorrhage after the allergen challenge. This migration of platelets occurred ahead of the infiltration of leukocytes into the lungs and was in single non-aggregated forms, which is different from the commonly observed aggregated state in thrombosis and hemostasis [38]. Due to the expression of chemokine receptors on the platelet surfaces, it is considered that they undergo chemotaxis in response to chemokines such as CCL11, CCL22, and CXCL12. Indeed, a recent study showed that CCR3 (CCL11 receptor) is essential for the recruitment of platelets into asthmatic lungs in the models of allergic inflammation. Under intravital microscopy, the rolling, adhesion, and extravascular migration of platelets in the HDM-sensitized murine models were markedly suppressed by SB328437, a CCR3 antagonist [36]. A study on platelet degranulation function in hemostasis and inflammation revealed that AHR and eosinophilic inflammation were significantly diminished in platelet-specific Munc13-4 KO mice, as the dense granule release was abrogated in these mice [45]. This evidence shows that platelets actively participate in the development of allergic asthma.

It is worth noting that while some have argued that allergens directly activate platelets based on the platelets’ anti-parasite cytotoxicity, others could not detect allergen-induced platelet aggregation and degranulation [39,46]. These contradicting results indicate that the underlying mechanism of platelets’ response to allergen is yet to be clarified. Interestingly, the work by Kasperska-Zajac, A. et al. found no significant differences in the plasma levels of PF-4 and β-TG in HDM-allergic patients and seasonal allergic rhinitis patients versus healthy non-atopic subjects [47,48]. Contrary to the notion that the vascular endothelial growth factor (VEGF) level is increased in plasma of allergic patients with asthma and atopic dermatitis, the work by Koczy-Baron, E. et al. showed that the free circulating VEGF level did not change in patients with chronic allergic rhinitis [49]. These discrepancies indicate that differences may exist in the platelet activity among patients with distinct clinical manifestations of atopy. This intriguing phenomenon was discussed in the review by Potaczek, D.P. [50].

## 3. Mechanisms of Platelets’ Role in the Pathogenesis of Allergic Asthma

It is now widely accepted that the intrapulmonary migration of platelets is a critical process for the development of allergic asthma because it not only promotes the recruitment of eosinophils and other leukocytes from the blood vessels into lung tissues but also serves as a reservoir of inflammatory mediators to promote the pathogenic process of allergic asthma [38]. In this section, we will discuss how intrapulmonary platelets contribute to the pathogenesis development of asthma. 

### 3.1. Platelets in Adaptive Immune Response Induction 

Type 2 immune response is recognized as an important mechanism in allergic diseases [51]. In recent years, the role of platelets in the induction of Type 2 immune response has attracted increasing attention. It is well established that dendritic cells (DCs) are the important antigen-presenting cells (APCs) which initiate and maintain Type 2 immune responses [52]. Amison, R.T. et al. demonstrated that platelets play a pivotal role in the sensitization stage of allergic asthma. They discovered that platelets rely on an IgE-FcεRI-dependent pathway to interact with pulmonary CD11c^+^ mononuclear cells (for example, CD11c^+^ DCs) which present antigen signals to T cells in bronchial lymph nodes and induce the formation of Type 2 immune response. When platelets were temporarily depleted during the sensitization period, the OVA-sensitized mice failed to develop an effective Type 2 immune response [53]. However, the underlying mechanism of the interaction between platelet and APC through the allergen-IgE-FcεRI axis remains obscure. On the other hand, elegant experiments by Dürk and colleagues ruled out the notion that platelets enhance the function of mature DCs in polarizing Th2 cells by secreting 5-HT and thus promoting the formation of Type 2 immunity [54,55]. It was reported that platelet activation promoted Th2 response and the development of asthma by upregulating the expression of CD40L (CD154) and partially inhibiting Foxp3^+^ regulatory T cell, resulting in a polarized Th2 response in allergic asthma [56]. Nakanishi, T. et al. put forward the idea that activated platelets participate in the Th2 immune response process in a CD154-independent fashion. They found that activated platelets could interact with thymic-stromal-lymphopoietin (TSLP)-stimulated myeloid DCs (TSLP-mDCs) by expressing RANK ligand (RANKL) to promote the maturation of TSLP-mDCs through the RANKL-RANK pathway. Mature TSLP-mDCs not only promoted the differentiation of naive T cells into Th2 cells but also drove the chemotaxis of memory Th2 cells to inflammatory sites by secreting chemokine CCL17, thereby maintaining the Th2-cell-mediated allergic condition [42].

Apart from Th2 cells, other mucosal innate immune cells were found to contribute to allergic asthma as well. Group 2 innate lymphoid cells (ILC2s), abundant in the lungs, intestines, and skin, serve as a source of type 2 cytokines alongside B and T cells [57]. Environmental allergens can damage the airway epithelial barrier and lead to the secretion of epithelial-derived alarmins such as IL-25, IL-33, and TSLP, which activate ILC2s. Takeda, T. et al. [58] confirmed that IL-33 is constitutively expressed on human platelets and megakaryocytes and could act as an essential regulator of eosinophilic inflammation in the airway. In the papain-induced murine airway eosinophilic inflammation model, depletion of platelets resulted in a remarkable decrease in the eosinophil counts in the BALF. Conversely, administration of wild-type platelets into IL-33–deficient mice significantly restored the eosinophil infiltration in an IL-33-dependent pathway. Based on these discoveries, they proposed that platelets might play an important role in airway Type 2 inflammation as an important cellular source of IL-33 [58]. In addition, platelets of asthmatic patients are thought to participate in tissue damage and repair via the Wnt signaling pathway. Chae, W.J. et al. [59] found that circulating Dickkopf-1 (DKK-1), which is mainly produced by platelets, inhibits the Wnt signaling pathway in type-2-cell-mediated inflammation. Upon the allergen challenge, the increase in circulating DKK-1 facilitated leukocyte migration and promoted Th2 cell differentiation and Type 2 cytokine production through the mTOR and MAPK signaling pathways [59].

### 3.2. Platelets in the Recruitment of Inflammatory Cells

Leukocyte infiltration in the airways during allergic asthma is initiated by chemokines from immune cell milieu or adhesion molecules expressed on cell surfaces. Platelets are involved in the pathogenesis of allergic asthma (Table 2). There are nearly 300 different proteins stored in platelet α-granules, such as CXCL1, PF4, β-TG, CXCL5, CXCL7, CXCL12, macrophage inflammatory protein-1α (MIP-1α), and RANTES (CCL5), which play an important role in leukocyte infiltration and inflammatory response [60]. In allergic diseases, PF4 induces the expression of IgG and IgE receptors, stimulates the release of histamine of basophils, activates eosinophils and other inflammatory cells, upregulates the expression of adhesion molecules, and facilitates adhesion between leukocytes and the endothelia, thus exacerbating allergic reactions [61]. RANTES (CCL5) binds to MIP-1α to recruit monocytes to the inflammatory site, attracts eosinophils and induces their degranulation and respiratory burst, promotes T cell activation and maturation, and recruits memory T cells to the lungs [62,63].

Among the adhesion molecules, P-selectin is the most well-studied glycoprotein and is found in α-granules of resting platelets and Weibel–Palade bodies of endothelial cells. It is a platelet activation surface maker induced by inflammatory or pro-aggregatory stimuli [66]. Platelets bind to eosinophils through the interaction between P-selectin and PSGL-1 and form platelet-leukocyte aggregates to mediate eosinophil migration into the lungs, thus triggering eosinophilic airway inflammation. This platelet and eosinophil interaction activates eosinophils to upregulate the expression of the integrins α4β1 (VLA-4) and αMβ2 (CD11b/CD18, Mac-1), which mediate the attachment of eosinophils to endothelia and thus promote recruitment of eosinophils to the inflammatory site [67]. 

Intriguingly, the formation of platelet-leukocyte aggregates influences the leukotriene synthesis in leukocytes as well. The role of cysteinyl leukotriene (CysLTs) and their receptors in the development of allergic asthma is highlighted in various studies. Activated platelets enhance the arachidonic acid metabolism, resulting in the increased production of CysLTs such as LTC_4_, LTD_4_, and LTE_4_. This process is considered to be an important mechanism in the pathogenesis of AERD [68]. AERD, accounting for a greater percentage of patients with severe asthma, is characterized by respiratory hyperreactions upon ingestion of COX-1 inhibitors and CysLTs overproduction, which are triggered by low doses of aspirin that inhibit COX-1 (cyclooxygenase-1) in platelets [69,70]. It represents a prototypical non-allergic endotype of asthma, along with a classic triad of symptoms (asthma, chronic rhinosinusitis with nasal polyposis, and hypersensitivity to aspirin and other cyclooxygenase-1 inhibitors) [71]. AERD is associated with increased CysLTs production and CysLTs receptor expression [72]. Elevated levels of platelet activation and the formation of platelet-leukocyte aggregates were observed in patients with AERD [69]. Leukocyte-adhering platelets convert the leukocyte-derived precursor leukotriene LTA_4_ to LTC_4_ through LTC_4_ synthase [68]. Prostaglandin E2 (PGE_2_), a downstream product of the COX-1 pathway which negatively regulates CysLT production by suppressing 5-lipoxygenase (5-LO) activity of leukocytes, was decreased by the COX-1 inhibitor and thus contributed to the CysLTs overproduction [72]. It is worth noting that platelets form aggregates with neutrophils and lymphocytes as well. Although allergic asthma is mostly eosinophilic, different degrees of neutrophil recruitment caused by allergen stimulation were observed in the airway of allergic asthma patients and asthmatic mouse models [73,74]. Activated platelets promoted the production of neutrophil superoxide anions in a P-selectin-dependent pathway [41]. The purine receptor P2Y1 on platelets also regulates leukocyte recruitment in allergic mice through the RhoA signaling pathway [65]. Moreover, myl9/12 secreted by platelets formed intravascular net-like structures that can bind to CD69^+^ CD4^+^ T cells [64].

### 3.3. The Roles of Platelets in Airway Hyperresponsiveness (AHR), Airway Remodeling, and Bronchoconstriction

Platelets can synthesize and release spasmogen, such as histamine, platelet activating factor (PAF), 5-HT, and thromboxane A2 (TxA2), that act on the airway smooth muscle cells to cause bronchoconstriction [75,76]. Depletion of platelets reduced allergen-induced airway hyperreactivity in allergic rabbits [77]. Studies on guinea pigs have shown that Bradykinin and capsaicin induced airways obstruction in a platelet-dependent manner [78]. PAF, a phospholipid derivative from various cells, including platelets, mast cells, basophils, eosinophils, and so on, is a strong activator of platelets and is well acknowledged for its role in bronchoconstriction, bronchial hyperresponsiveness, mucus hypersecretion, and gas exchange impairment [79]. TxA2, a potent airway smooth muscle contractile agent predominantly produced by platelets, is associated with airway inflammation and bronchial hyperreactivity [76]. Platelets act as a major reservoir and means of transport for brain-derived neurotrophic factor (BDNF). BDNF was found to contribute to airway obstruction and hyperresponsiveness in a model of allergic asthma [80]. Increased BDNF concentrations in platelets of asthmatic patients were found to correlate with the clinical parameters of airway dysfunction [32]. Recently, it was reported that the sphingolipid metabolism was altered in the patients allergic to house dust mites. Activated platelets are rich sources of sphingosine-1-phosphate (S1P), which acts directly on the smooth muscle cells to promote proliferation and AHR. The severity of allergen-induced bronchoconstriction is highly correlated with the plasma concentration of S1P [33].

Platelets are also rich in proteases, mitogens, and growth factors, which not only participate in the damage and repair process of allergic asthma, but also affect the phenotypes of airway epithelial cells, fibroblasts, and airway smooth muscle cells [81]. Metalloproteases (e.g., MMP2, MMP9), free radicals, and cationic proteins can degrade extracellular matrix, increase the vascular permeability of airway epithelia, and stimulate mucus secretion [60]. In murine models of chronic allergic inflammation, platelets are indispensable for the structural remodeling of the airway after chronic exposure to aerosolized allergens [23]. Platelet-derived growth factor (PDGF) promotes the proliferation of human airway smooth muscle cells as a mitogen, but is also involved in airway fibrosis and airway remodeling as a strong chemokine of fibroblasts [82,83,84]. In a mouse model with repeated exposure to allergens, PDGF overexpression and airway smooth muscle thickening were observed [85]. Vascular endothelial growth factor (VEGF) induces endothelial cell growth and angiogenesis and increases vascular permeability, and platelets secrete VEGF upon activation in vivo [86]. A study showed that VEGF levels are associated with the degree of vascularity and are inversely correlated with the airway caliber and level of AHR [87]. In allergic asthma, platelets inhibit eosinophil apoptosis by secreting granulocyte-macrophage colony-stimulating factor (GM-CSF), thus extending chronic inflammatory response and tissue damage [40]. In addition, GM-CSF enhances the 5-lipoxygenase (5-LO) activity in neutrophils and eosinophils and promotes the production of CysLTs, causing airway smooth muscle contraction, inflammatory cell recruitment, and tissue edema [68].

In conclusion, platelets interact with multiple inflammatory cells vital in the inflammatory process and development of allergic asthma occurs through direct cell-to-cell contact or inflammatory mediators, thus deeply involved in the pathogenesis of allergic asthma (Figure 2).

## 4. Antiplatelet Therapies for Asthma Control

Currently, the treatments for allergic asthma include various strategies, including anti-inflammatory agents, bronchodilators, allergen-specific immunotherapy, and biologics targeting eosinophil activation and cytokines production, such as anti-IL-5 therapy. However, these strategies often do not result in full resolution for most asthmatic endotypes [88]. Hence, a new therapy to treat allergic asthma is urgently needed. Given that platelets are extensively involved in the pathogenesis of allergic asthma, it is possible in principle to treat allergic asthma with antiplatelet drugs (Table 3). Common antiplatelet drugs include ADP receptor antagonists, thromboxane synthase inhibitors, thromboxane-prostanoid receptor (TP receptor) antagonists, 5-HT modifier, and cyclooxygenase (COX) inhibitors.

Platelets express four different purinergic receptors, P2Y1, P2Y12, P2Y14, and P2X1, on their surface and they play critical roles in hemostasis, thrombosis, and inflammation [98]. The G-protein-couple receptors (GPCRs) P2Y1 and P2Y12, when stimulated by ADP, induce platelet activation and aggregation. P2X1 is a ligand-gated ion channel and induces calcium mobilization and platelet shape change when activated by ATP [99]. The pyrimidine nucleotide-sensitive P2Y14 is a newly found P2 receptor whose function is still unknown [100]. Mechanistically, purine and pyrimidine nucleotides, released by damaged cells from inflammatory, traumatic, and ischemic conditions, interact with P2 receptors on platelets to assist in stabilizing platelet aggregation as well as recruiting and activating leukocytes. Blocking these receptors may prevent adverse cardiovascular events as well as secretion of proinflammatory mediators [101]. Clopidogrel, Prasugrel, and Ticagrelor are commonly used selective P2Y12 receptor antagonists and antithrombotic agents for various cardiovascular diseases [102,103]. P2Y12 receptor antagonists can also limit the release of platelets and the formation of platelet-leukocyte aggregates [89,90]. Clopidogrel inhibited eosinophilic inflammation and airway hyperreactivity in OVA-sensitized mice [91]. Combined use of Clopidogrel and Montelukast had a synergistic effect in asthmatic treatment [104]. However, Clopidogrel and other P2Y12 receptor antagonists, such as MRS2395 and AR-C66096, did not inhibit the infiltration of leukocytes in pulmonary inflammatory responses [65,105]. A proof-of-concept, randomized, controlled study found that Prasugrel only slightly reduced the burden of bronchial inflammation [92]. Therefore, larger-scale clinical studies are needed for P2Y12 receptor antagonists in the treatment of allergic asthma. Platelet P2Y1 receptors also participate in the pathological process of allergic asthma. Different ligands activate different P2Y1 receptor signaling pathways to promote inflammation by binding to different sites of P2Y1 receptors. The selective P2Y1 receptor antagonists MRS2179 and MRS2500 inhibited the recruitment of eosinophils and lymphocytes in the lungs of allergic inflammatory mice [65]. Therefore, the crystal structure of P2Y1 receptor is of great significance for designing a new generation of specific P2Y1 receptor antagonists to block the initiation or amplification of inflammatory signaling pathways [106]. Although the mechanisms of purinergic receptors in allergic asthma require further study, they are potential therapeutic targets for allergic asthma.

TxA2, a lipid metabolite produced by activated platelets through the arachidonic acid metabolic pathway, is an important proinflammatory mediator. In allergic asthma, TxA2 causes bronchial contraction, increased vascular permeability, tissue edema, and airway hyperresponsiveness [107]. Blocking TxA2 synthesis reduces the concentration of TxA2 during asthma attacks, but another bronchoconstrictor, prostaglandin, which acts together with TxA2 by binding to the TxA2 receptor (TP receptors), is generated due to the diverted arachidonic acid metabolism pathway [108]. Therefore, the combined use of TxA2 synthase inhibitor and TP receptor antagonist could be more effective in the treatment of allergic asthma. Indeed, the administration of Ozagrel (OKY-046, a selective TxA2 synthase inhibitor) and S-1452 (a TP receptor antagonist) inhibited the production of proinflammatory cytokines and prevented the infiltration of eosinophils into the airway [93]. Similarly, Seratrodast (AA-2414), a TP receptor antagonist, was reportedly able to reduce bronchial hyperresponsiveness by reducing airway inflammation [95]. ONO-1301, a novel prostacyclin agonist and TxA2 synthase inhibitor, was shown to suppress AHR and airway inflammation in asthma [94]. While TxA2 synthetase inhibitors and TP receptor antagonists were found to have positive effects on allergic asthma in Japan [109], studies from Western countries revealed no obvious curative effects. This discrepancy may be associated with the genetic polymorphism in TxA2 synthetase and TP receptors [108].

It has been demonstrated that platelets are an important source of 5-HT in peripheral blood. 5-HT is stored in the δ-granules of platelets after the uptake from intestinal enterochromaffin cells (ECs) and is released after platelet activation [110,111]. In patients with allergic asthma, the concentration of 5-HT in the lungs is closely correlated with the concentration of platelets in the blood. The increased concentration of 5-HT in the plasma is observed in patients with symptomatic asthma, and the level of free 5-HT is closely related to disease severity and pulmonary function [112]. Tianeptine enhances the uptake of free 5-HT by 5-HT-specific transporter in the resting platelets in peripheral blood for storage in δ-granules via 5-HT-specific transporters [96,113]. Tianeptine was effective in controlling asthma in two double-blind placebo cross-trials and in an open study covering 25,000 asthma patients over seven years [97].

It is noteworthy that aspirin, a member of NSAIDs that is commonly used to prevent thrombosis, is a selective COX-1 inhibitor which plays an antithrombotic role by inhibiting TxA2 synthesis in platelets [114]. Clinical studies have revealed that some patients with allergic asthma show intolerance to aspirin and may even develop AERD, mainly due to the significantly reduced PGE_2_ synthesis. Mast cell activation and the interaction between platelets and granulocytes lead to overexpression of CysLTs. All of these factors contribute to constriction of the bronchioles, acute exacerbation of asthma, and increased mucus production in AERD [115,116]. Therefore, aspirin should be used with caution when treating allergic asthma. Presently, AERD treatments include intravenous corticosteroids, anti-IgE, anti-IL-4/13, aspirin desensitization, and leukotriene-modifying drugs (leukotriene receptor antagonists and 5-lipooxygenase inhibitors) [70,71]. Due to the elevated levels of platelet activation and platelet-leukocyte aggregates in AERD, anti-platelet therapy would be an efficacious treatment, although clinical trials exploring the potential for platelet-targeted therapies in AERD are still in the early stages [11].

The application of antiplatelet drugs to allergic asthma needs to be further studied. Page CP pointed out in 1988 that platelets use different signaling pathways during hemostatic and inflammatory response. In an inflammatory response, inflammatory factors evoke platelet activation, and platelets participate in the inflammatory response through adhesion and secretion activities, but this generally does not include the aggregatory response [117]. Consistent with this notion, clinical studies found minor coagulation defects and prolonged bleeding time in patients with allergic asthma. Therefore, it is of great value to further study the underlying mechanism of platelets in the inflammatory process, which can provide a new direction for new therapeutics for allergic asthma.

## 5. New Insights of Platelets in Asthma

The complement system, an important part of innate immunity which acts quickly on the site of inflammation, has been found to play a role in the development of allergic asthma, especially the anaphylatoxins (C3a and 5a) [118]. For example, a study showed that C3 activates platelets to release 5-HT [119]. Growing evidence supports the hypothesis that the complement system connects inflammation and thrombosis [120,121]. Moreover, platelet activation leads to complement activation through different mechanisms, such as the binding of P-selectin with C3b [122], indicating a possible interaction between platelet and complement system in allergic asthma. However, studies on the relationship between the complement system and platelets are limited, so it will be very interesting to further explore this relationship.

Similarly, alveolar macrophages (AMs), residing in the epithelial surface of airways and lungs, are considered to be the key sensors and effectors for the changing environment [123]. Macrophage polarization is found to have a profound impact on asthma pathogenesis. Increased polarization and activation of M2 macrophage promote the pathogenesis of allergic asthma [124]. The interaction between platelets and macrophages has not been extensively explored so far. A recent article revealed that in bacteria-induced pulmonary inflammation, platelets educate AMs toward an anti-inflammatory phenotype (featured as CD68^+^ CD163^+^ AMs) that mediates remission [125].

Numerous studies have shown that mitochondrial dysfunction is an active contributor to the pathological condition and drives disease progression of asthma. The oxygen-rich environment and large contact areas of the alveoli render blood components, including platelets, sensitive to oxidative stress and damage [126]. Compared with healthy subjects, platelets from asthmatic patients are less dependent on glycolysis but are more dependent on TCA cycle which was evidenced by the increased activity of Krebs cycle enzymes [127]. Since the bioenergetics of platelets in asthmatic patients are altered, it was proposed that bioenergetics of circulating platelets could mirror those of airway epithelia in healthy and asthmatic individuals [126]. Thus, platelets are potential markers to monitor bioenergetic changes in asthmatic patients.

Up to now, research on the relationship between platelets and asthma mainly focused on Type 2 immunity. However, it is now generally believed that Type 2 immune-mediated response alone cannot explain the heterogeneity of asthma. Asthmatic patients showed increased neutrophils and Th17 cytokines in the BALFs [128], and platelet-derived mediators, such as CXCL1, CXCL4, and CCL5, promote the differentiation of Th17 cells [129,130]. Furthermore, platelets differentially regulate proliferation of CD4^+^ effector T cells (Th1, Th17, and Treg) and thus induce distinct dynamics of immune response [131].

The discovery of megakaryocytes (MKs), the precursors of platelets, in the lungs further confirmed the role of platelets in inflammation [132]. An article that was published in *Nature* in 2017 [133] first provided direct evidence that the lung acts as a major site of platelet biogenesis and as storage for hematopoietic progenitors. From their work, they visualized large numbers of megakaryocytes circulating through the lungs. These megakaryocytes migrated from extrapulmonary sites such as bone marrow and accounted for nearly 50% of the total platelet production. In addition, through RNA-seq analysis, they found that lung megakaryocytes were prone to an innate immunity function [133]. Yeung, A.K. et al. recently provided further evidence for this finding [134]. Resident lung MKs express higher levels of immune molecules compared to those in the bone marrow, supporting the notion that lung MKs have similar gene expression patterns as APCs. For example, lung MKs expressed higher levels of TLR2, TLR4, and MHC-II-associated molecules, which are involved in microbial surveillance and antigen presentation [134]. Lung MKs are also able to induce CD4^+^ T cells activation and modulate immune response in an MHC-II-dependent manner [135]. These data suggest that the platelets-megakaryocytes system may act as an integral part of the host defense system.

The mechanism by which platelets contribute to the development of allergic asthma is mainly conducted on the mouse models due to the similarity in the coagulation system between mouse and human [136]. However, there are also clear differences in the immune system between mouse and human [137,138]. Thus, developing humanized mouse models could serve as an important preclinical tool for asthma research.

## 6. Conclusions

Allergic asthma involves a variety of inflammatory cells and factors and its pathogenesis is complex. Platelets are anucleate cells and their numbers are more than 10 times that of white blood cells. They carry a variety of granules and special structures like open canalicular system (OCS) and dense tubular system (DTS). As a rich reservoir of inflammatory factors, they are highly reactive and secretory, and are of significant importance in the immune system. In allergic asthma, platelets can not only facilitate leukocytes to migrate into the lung, but also polarize adaptive immune response through a variety of possible ways: they promote differentiation and activation of Th2 and ILC2s, promote IgM to IgE switch, inhibit apoptosis of eosinophils in the lung, and induce allergy to innocuous allergens in the environment. Though a large number of studies suggest that platelets are involved in the development of allergic asthma, the exact molecular mechanisms are still not fully understood, especially the complex processes of allergen-specific activation of platelets and the induction of adaptive immune response as well as the development of AHR. Furthermore, the bioenergetic changes in platelets in asthma and the finding of platelet precursors, megakaryocytes, in lungs suggest the importance of studying the coagulation and hemostasis system in asthma. A powerful method for studying the pulmonary immune environment is now available with the advent of intravital imaging [139]. Therefore, exploring the role of platelets in the development of allergic asthma will not only enable us to have a more comprehensive understanding of the pathogenesis of allergic asthma, but will also provide new and better treatments for allergic asthma.

## Figures and Tables

**Figure 1 cells-10-02038-f001:**
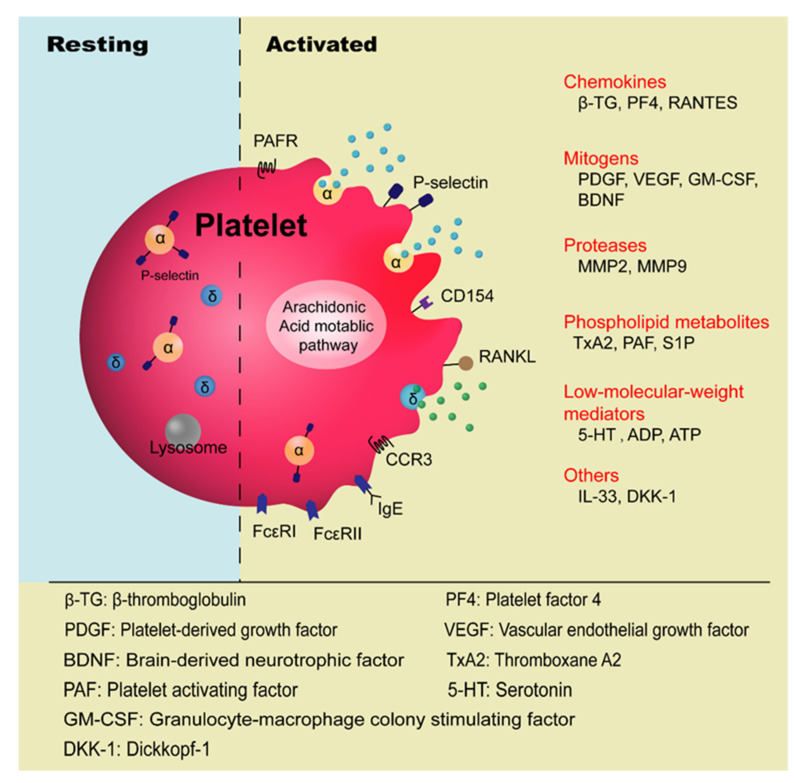
Platelet-derived factors contributing to allergic asthma.

**Figure 2 cells-10-02038-f002:**
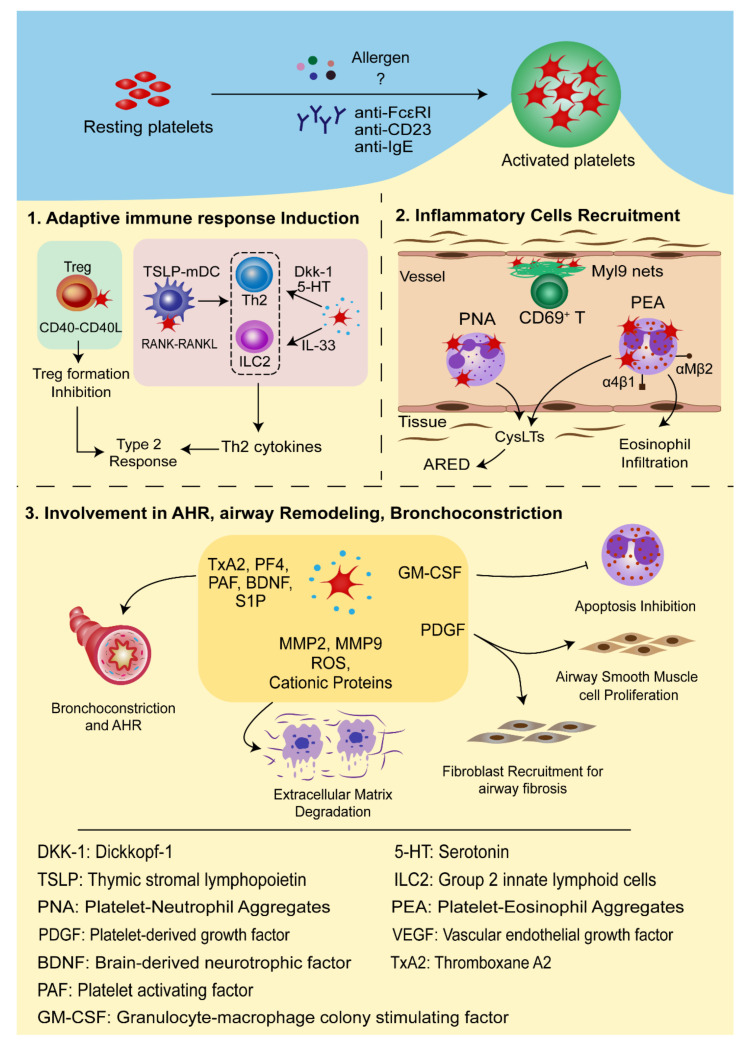
Involvement of platelets in the pathogenesis of allergic asthma. Environmental allergens, anti-IgE, and/or anti-IgE receptors (anti-FcεRI and anti-FcεRII/CD23) activate resting platelets. (1) Activated platelets secrete IL-33 that acts on ILC2, or Dkk-1 and 5-HT to promote naive T helper cells’ differentiation into Th2 effectors. Moreover, they can influence the activity of TSLP-stimulated mDCs in a RANK-RANKL-dependent pathway or inhibit regulatory T cells’ differentiation by overexpressing CD40L (CD154). (2) Activated platelets are involved in the recruitment of inflammatory cells. Platelets form aggregates with leukocytes through P-selectin-PSGL-1 interaction after allergens stimulation. The platelet-eosinophil aggregates (PEA) activate eosinophils and upregulate the expression of αMβ2 and α4β1 integrins, through which PEA can adhere to the endothelial cells and migrate to the inflammatory sites. The platelet-neutrophil aggregates (PNA) activate the arachidonic acid pathway to generate CysLTs which contribute to ARED. The CD69-Myl9 system is a newly found mechanism mediating the infiltration of activated CD4^+^CD69^+^ T cells into the inflamed tissue. (3) Activated platelets produce spasmogen, such as TxA2, PF4, PAF, BDNF, and S1P, to induce bronchial smooth muscle contraction as well as airway hyperresponsiveness. They also release MMP2, MMP9, cationic proteins, and ROS to cause degradation of extracellular matrix. PDGF promotes proliferation of fibroblasts and airway smooth muscle cells. GM-CSF inhibits apoptosis of eosinophils and thus contributes to the chronic inflammatory response and tissue damage. GM-CSF also enhances the 5-LO activity in neutrophils and eosinophils to accelerate the production of CysLTs, causing airway smooth muscle contraction, inflammatory cell recruitment, and edema. All these processes together lead to chronic airway inflammation and airway remodeling.

**Table 1 cells-10-02038-t001:** Various platelet-derived factors in the pathogenesis of allergic asthma in humans.

Models	Samples	Indicators	Subjects	References
In vivo	Peripheral blood	Platelet-leukocyte conjugates↑	Asthmatic patients after allergen exposure	[25]
		Platelet-derived microparticles (PMPs) ↑	Asthmatic patients	[30]
		Percentage of IgE^+^ platelets↑	Asthmatic patients	[31]
		Eosinophil β1-integrin activation↑	Asthmatic patients	[28,29]
		Platelet BDNF↑	Patients with allergic asthma	[32]
	Plasma	β-TG and PF4↑	Atopic dermatitis patients with concomitant asthma and allergic rhinitis	[19]
		β-TG, PF4↑ Platelet count↓	House-dust-mite-sensitive asthmatic patients intrabronchially challenged with Dp extract	[20]
		PF4↑ (during the grass pollen season) PF4↓(off season)	Patients with pollen-induced seasonal allergic rhinitis and asthma	[26]
		BDNF↑	Patients with allergic asthma	[32]
		Phingosine-1-phosphate (SIP)↑	House-dust-mite-allergic patients	[33]
	BALF	Isolation of platelets	Asthmatic patients	[21]
		5-HT↑	Asthmatic patients	[31,34,35]
		β-TG, PF4↑	Ragweed-allergic asthmatic subjects after challenge with ragweed antigen.	[22]
	Bronchial biopsies	Platelet deposition on interalveolar septum walls Platelet number↑	Asthmatic patients	[23,24,36]
	Nasal lavage fluid	P-selectin positively correlated with ECP level	Asthmatic patients	[27]
In vitro	Platelet	FcεRI and FcεRII/CD23 expression	Human platelets and megakaryocyte	[31,34,35,37,38]
		RANTES release↑ Cytotoxicity against schistosomula	Platelets treated by anti-FcεRI, anti-CD23, anti-IgE	[34,35,37]
		RANTES release↑	Platelet from allergic patients stimulated with IgE and anti-IgE	[35]
		Allergen-specific cytotoxicity against schistosomula↑	Patients allergic to *Dermatophagoides pteronyssinus*	[39]
		Allergen-specific platelet chemotaxis	Allergic asthmatic patients	[38]
		GM-CSF↑ Eosinophils apoptosis↓	Human platelets and Eosinophils coculture	[40]
		P-selectin↑ Neutrophil superoxide anions↑	Human platelets and neutrophils coculture	[41]
		RANKL in platelets↑ CCL17 (Th2-attracting chemokine)↑	TRAP6-activated platelets with TSLP-stimulated DCs coculture	[42]

**Table 2 cells-10-02038-t002:** Evidence of platelets’ involvement in the pathogenesis of allergic asthma.

Mechanism Studied	Method/Animal Model	Findings	References
Platelet degranulation	Munc13-4−/−mice	Platelets Munc13-4 deficiency abolishes dense granule release to reduce AHR and eosinophilic inflammation.	[45]
Platelet migration	FcRγ−/−mice Platelet depletion Platelet transfusion Allergen or anti-IgE antibody in vitro stimulation	Platelet migration is allergen-IgE-FcεRI dependent.	[38]
	CCR3 antagonist	Platelet rolling, adhesion, and extravascular migration are CCR3 dependent.	[36]
Sensitization of allergic asthma	Platelet depletion FcRγ−/−mice	Platelets rely on an IgE-FcεRI-dependent pathway to induce Type 2 immune response formation during sensitization stage.	[53]
Leukocyte recruitment	Platelet depletion Platelet transfusion	Depletion of platelets reduces, while transfusion of platelets restores the allergen-induced pulmonary leukocyte recruitment.	[25]
	Platelet depletion Platelet transfusion	Platelets P-selectin is required for pulmonary eosinophils and lymphocytes recruitment.	[43]
	Selectin−/−mice	P-selectin is critical in the development of allergen-induced airway response.	[43,44]
	Anti-Myl9/12 antibody CD69-deficient mice	Platelet secreted myl9 upon activation and formed intravascular net-like structures to bind to CD69^+^ CD4^+^ T cells, regulating the pathological process of allergic asthma.	[64]
	P2Y_1_ antagonists Platelet depletion Platelet transfusion	Purine receptor P2Y_1_ on platelets regulates leukocyte recruitment in allergic mice through the RhoA signaling pathway.	[65]
Type 2 immunity induction	TPH1−/−mice BM chimera Mast cell–deficient mice Platelet transfusion	Platelets secrete 5-HT to enhance the ability of mature DCs to polarize Type 2 immunity formation.	[55]
	Platelet depletion Platelet transfusion Cd154−/−mice	Platelets inhibit Treg generation via CD154 to promote asthma development.	[56]
	Platelet depletion Platelet transfusion IL-33-deficient mice	IL-33 is essential for eosinophilic inflammation in the airway.	[58]
	Dkk1^d/d^ mice	Dickkopf-1 (DKK-1) facilitated leukocyte migration and promoted Th2 cell differentiation and Type 2 cytokine production.	[59]
Airway remodeling	Platelet depletion	Platelets are necessary for epithelial and smooth-muscle thickening and the deposition of reticular fibers in the extracellular matrix (ECM).	[23]

**Table 3 cells-10-02038-t003:** Antiplatelet drugs for treating allergic asthma.

Category	Target	Drug	Mechanism	References
ADP receptor antagonists	P2Y12 receptor	Clopidogrel Prasugrel Ticagrelor	Reduce the release of platelets and the formation of platelets-leukocyte aggregates, and inhibit eosinophilic inflammation and airway hyperreactivity.	[89,90,91,92]
	P2Y1 receptor	MRS2179 MRS2500	Inhibit the recruitment of eosinophils and lymphocytes to the lung.	[65]
TxA2 synthase inhibitor	TxA2 synthase	Ozagrel (OKY-46) ONO-1301	Inhibit the production of proinflammatory cytokines and alleviate the eosinophil infiltration in the airways; suppress AHR and airway inflammation.	[93,94]
TP receptor antagonist	TP receptor	Seratrodast (AA-2414) S-1452	Reduce bronchial hyperresponsiveness by reducing airway inflammation.	[93,95]
5-HT modifier	5-HT-specific transporter	Tianeptine	Enhance the uptake of free 5-HT in peripheral blood.	[96,97]

## Data Availability

Not applicable.

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
