# Peer review of "Platelets, Not an Insignificant Player in Development of Allergic Asthma"

_cells, 2021, doi:10.3390/cells10082038_

Round 1

Reviewer 1 Report

The review by Luo and colleagues reports on published findings on platelets in allergic asthma, regarding evidence of involvment, pathogenesis, and therapy. It is a quite rarely treated topic despite its potential major interest. Ly comments are the followings.

Major Comments.

  1. The three main sections are not really reflecting their content. The reviewer proposed the followings: (1) descriptive findings, (2) potential mechanisms of involvment, (3) therapeutic implications. The authors should, as far as possible, precise the level of evidence when referring to their statements about relationships between platelets and asthma; association does not mean causality.
  2. A table or figure on platelets, summarizing their main features (content, functions) would be helpful.
  3. If sufficient data are available, it would be indicated to make one paragraph on the involvment of platelets in non-allergic asthma. Alternatively, the authors should consider discussing the relationship between platelets and asthma irrespectively of allergy.
  4. Releted to the previous comment is a note concerning apsirin-induced asthma. The paragraph treating this aspect should be revised whilst considering the pathophysiology of this entity, which is related to a pharmacological susceptibility to NSAID and aspirin due to genetic polymorphisms of COX genes, irrespectively of allergy. This phenotype (called AERD) is a prototypical non-allergic endotype of asthma, sometimes associated with nasal polyposis (hence called Samter's or Widal's triad, i.e. asthma, polyposis and aspirin hypersensitivity).

Minor comments.

  1. The reviewer suggests to use the term 'airway' instead of "pulmonary" when speaking on fibrosis (e.g. figure 1), to avoid confusion with true pulmonary fibrosis related to the lung interstitium. Effects on airway vs alveolar fibroblasts should be distinguished.
  2. Some English revision would be ideal.

Author Response

Major Comments.

Point 1: The three main sections are not really reflecting their content. The reviewer proposed the followings: (1) descriptive findings, (2) potential mechanisms of involvement, (3) therapeutic implications. The authors should, as far as possible, precise the level of evidence when referring to their statements about relationships between platelets and asthma; association does not mean causality.

Response 1: We revised the subtitles of each section following as i) Current Understanding of Platelets’ Involvement in Allergic Asthma (page 2 line 71-72) ii) Mechanisms of Platelets’ Role in the Pathogenesis of Allergic Asthma (page 4 line 179) iii) Antiplatelet Therapies for the Asthma Control (page 13 lines 383-384)

To precisely describe the platelet roles in the pathogenesis of allergic asthma in human and in animal models, respectively, we added on Tables 1 & 2 in Pages 11 to 13. We divided the statements/evidence into 3 levels based on the strength of the evidence: 1) evidence of platelet activation in vivo in asthma patients. 2) human platelets activation or interaction with other human cells in vitro, 3) involvement of platelets in animal asthmatic models.

Point 2: A table or figure on platelets, summarizing their main features (content, functions) would be helpful.

Response 2: We added Figure 1 on Page 8 titled “Platelet-derived factors contributing to allergic asthma”.

Point 3: If sufficient data are available, it would be indicated to make one paragraph on the involvement of platelets in non-allergic asthma. Alternatively, the authors should consider discussing the relationship between platelets and asthma irrespectively of allergy.

Response 3: While we focused on platelet involvement in allergic asthma in this review, we added more information on AERD, the most common type of non-allergic asthma, in Page 6 as (4#) suggested.

Point 4: Related to the previous comment is a note concerning apsirin-induced asthma. The paragraph treating this aspect should be revised whilst considering the pathophysiology of this entity, which is related to a pharmacological susceptibility to NSAID and aspirin due to genetic polymorphisms of COX genes, irrespectively of allergy. This phenotype (called AERD) is a prototypical non-allergic endotype of asthma, sometimes associated with nasal polyposis (hence called Samter's or Widal's triad, i.e. asthma, polyposis and aspirin hypersensitivity).

Response 4: We summarized the available data on the platelets’ role in AERD in Page 6 Lines 266-279.

Minor comments

Point 1: The reviewer suggests to use the term 'airway' instead of "pulmonary" when speaking on fibrosis (e.g. figure 1), to avoid confusion with true pulmonary fibrosis related to the lung interstitium. Effects on airway vs alveolar fibroblasts should be distinguished.

Response 1: We revised it accordingly.

Page 10 Figure 2 “Fibroblast recruitment for pulmonary fibrosis” has been corrected as “Fibroblast recruitment for airway fibrosis”.

Page 7 Line 325 “pulmonary airway fibrosis and airway remodeling as a strong chemokine of fibroblasts”.

Point 2: Some English revision would be ideal.

Response 2: We polished the language in the revised manuscript.

Reviewer 2 Report

The authors have reviewed the function of platelets in the lungs and collected information on their role in IgE-mediated pathways and diseases. The topic is interesting and a lot of new data is gathered during the last few years, therefore this is a good topic for a review.

- There's a valuable and recent (2017) article published in Nature (Nature. 2017 April 06; 544(7648): 105–109. doi:10.1038/nature21706.), please use that as a reference and bring that information into your paper.

-If you introduce allergic asthma, you should introduce also AERD, respectively.

-Remove lines 58-61, there’s no new information there.

-Line 98, a reference is missing

-Is there any interaction of platelets with complement factors, since both are quickly on the site of inflammation.

-Line 134-135, a reference is missing

-How platelet-IgE-RceRI are interacting with APC?

-Usually, alveolar macrophages are important in the induction of immune responses in the lungs. How are alv MOs related to platelets?

-Line 136-137: …APCs which transmit antigen signals… Correct to “…APCs which present antigens…”

- You tell quite a lot of results obtained from rabbits and from guinea pigs. How close they are when compared to humans? Are their platelets similar?

-Figure 1: enlarge the text, now it’s very small especially at the part 2 (dark background under the “PEA”).

-The first paragraph of chapter 4 (Antiplatelet Therapy and the Control of Asthma (lines 279-288): This whole chapter is full of details and floating information. Please, make a table or a

fig to show, what is the real function of each R/mediator/cytokine, and how its blocking changes some responses.

-Please explain at mechanistical level, how the purinergic receptors function during normal responses (lines 289 ->), and after that how the antagonists change those functions (at mechanistic level).

-Please add the full name for each abbreviations when mentione fore the first time, for example TxA2 at the line 315?

Author Response

Point 1: There's a valuable and recent article published in Nature (Nature. 2017 April 06; 544(7648): 105–109. doi:10.1038/nature21706.), please use that as a reference and bring that information into your paper.

Response 1: We incorporated the article in Pages 16-17 Lines 541-548 as suggested.

Point 2: If you introduce allergic asthma, you should introduce also AERD, respectively.

Response 2: We summarized the available data about the platelet in AERD in Page 6 Lines 266-279.

Point 3: Remove Lines 58-61, there’s no new information there.

Response 3: We have deleted the repetitive contents. Page 2 Lines 60-64

Point 4: Line 98, a reference is missing

Response 4: We added a reference in Page 3 Lines 131-132 “Pitchford SC, et al. [35] They showed that in OVA-sensitized mice, platelets migrated out of the blood vessels”

Point 5: Is there any interaction of platelets with complement factors, since both are quickly on the site of inflammation.

Response 5: The complement system acts as an innate immune system and reacts quickly on the site of inflammation. It plays a role in the development of allergic asthma, especially the anaphylatoxins (C3a and 5a). We added more information in Page 16 Lines 501-511 to describe more extensively on the roles of platelets in allergic asthma.

Point 6: Lines 134-135, a reference is missing

Response 6: A reference was added in Page 4 Line 192: “Amison RT., et al [48]”.

Point 7: How platelet-IgE-RceRI are interacting with APC?

Response 7: To provide readers more accurate understanding of this aspect, we revised our statement in Pages 4-5 Lines 192-200.

Point 8: Usually, alveolar macrophages are important in the induction of immune responses in the lungs. How are alv MOs related to platelets?

Response 8: Alveolar macrophages are recognized as key sensors and effectors of the environment. There are few papers on the interaction between platelet and alveolar macrophage in the development of allergic asthma, even though the platelet involvement is intensively studied in the acute lung injury. However, we found and added an interesting paper demonstrating platelets orchestrate the polarization of alveolar macrophages in bacteria-induced pulmonary inflammation in Page 16 Lines 512-520.

Point 9: Lines 136-137: …APCs which transmit antigen signals… Correct to “…APCs which present antigens…”

Response 9: We corrected the error. See Page 4 Line 195.

Point 10: You tell quite a lot of results obtained from rabbits and from guinea pigs. How close they are when compared to humans? Are their platelets similar?

Response 10: Platelets play an important and similarity role in hemostasis and thrombosis in mammals, including human, mouse, rabbits, guinea pigs, etc, but with some differences. For example, platelets from different species respond differently to the physiological stimulus such as thrombin, ADP and PAF (PMID: 15921723, PMID: 12354523, PMID: 15714752). In contrast to human, guinea pig platelets displayed a stronger maximal response to ADP than to thrombin. EC50 for thrombin and ADP in mouse platelet were significantly lower compared to those in human. We proposed a possible difference that can explain the controversial results between human and animal models. See Page 17 Lines 559-565.

Point 11: Figure 1: enlarge the text, now it’s very small especially at the part 2 (dark background under the “PEA”).

Response 11: We corrected as suggested. See Page 10 Figure 2.

Point 12: The first paragraph of chapter 4 (Antiplatelet Therapy and the Control of Asthma (Lines 279-288): This whole chapter is full of details and floating information. Please, make a table or a fig to show, what is the real function of each R/mediator/cytokine, and how its blocking changes some responses.

Response 12: We made a table to summarize the clinical use of anti-platelet drugs as suggested. See Pages 15-16 Table 3.

Point 13: Please explain at mechanistical level, how the purinergic receptors function during normal responses (Lines 289 ->), and after that how the antagonists change those functions (at mechanistic level).

Response 13: We added the information in Pages 13-14 Lines 398-410.

Point 14: Please add the full name for each abbreviations when mentioned for the first time, for example TxA2 at the Line 315?

Response 14: We corrected the mistakes in Page 6 Lines 293-294. “Platelets can synthesize and release spasmogens, such as histamine, platelet activating factor (PAF), 5-HT and thromboxane A2 (TxA2),”

Reviewer 3 Report

In this review article, the authors summarize current knowledge of the roles of platelets in allergic asthma. Overall, this manuscript is well-written with appropriate citations, including the latest reports on this topic. Therefore, it would be of particular interest to the Cells readers and will help share up-to-date knowledge on this topic.

Author Response

Point 1: In this review article, the authors summarize current knowledge of the roles of platelets in allergic asthma. Overall, this manuscript is well-written with appropriate citations, including the latest reports on this topic. Therefore, it would be of particular interest to the Cells readers and will help share up-to-date knowledge on this topic.

Response 1: We appreciate the comments.

Reviewer 4 Report

With interest, I read the manuscript cells-1297967. Indeed, there is an unmet need for a review on the role of platelets in asthma, including potential platelet-related therapeutic approaches.

This manuscript cells-1297967 could be a good candidate for such a review. However, several major things would need to be addressed.

  1. The selection of the literature should be more comprehensive. Thus any wider (systematic?) literature search would be welcome. I mean, the literature in the field is not that abundant so including most of it if not mall should not be a problem.
  2. From my own knowledge and experience, I can tell you already that the crucial papers from at least three groups/sources have been fully or almost fully ignored, such as:
    1. PMID: 18797178,
    2. PMID: 21840179, 24315352, 30053974,
    3. PMID: 16297143, 27127523, 18193345.

All those papers should be incorporated independently of the further literature search.

  1. What is the chapter „2. Role of Platelets in Allergic Asthma“ for? Its title does not differ much from the title of the next section „3. Platelet Contribution in the Pathogenesis of Allergic Asthma“. Should chapter 2 not belong to section 3 as its subsection?
  2. Two additional comprehensive tables one summarizing human and the other animal most important mechanistic studies on the role of platelets in asthma should be added to make this review much more comprehensive.

Additional comments:

  1. Please, make sure that absolutely all abbreviations used in figures or tables are explained in their legends.

Author Response

Point 1: The selection of the literature should be more comprehensive. Thus any wider (systematic?) literature search would be welcome. I mean, the literature in the field is not that abundant so including most of it if not mall should not be a problem.

Response 1: We tried our best to include all important papers in the revised manuscript. To see Page 2 Lines 73-81, Pages 3-4 Lines 146-151, Page 4 Lines 156-166.

Point 2: From my own knowledge and experience, I can tell you already that the crucial papers from at least three groups/sources have been fully or almost fully ignored, such as:

PMID: 18797178,

PMID: 21840179, 24315352, 30053974,

PMID: 16297143, 27127523, 18193345.

Response 2: We inserted the suggested papers as suggested. To see Page 2 Line 47 Ref 10 (18797178); Page 2 Lines 74-81, Refs 17, 19 (21840179, 30053974), Page 4 Lines 156-166 Refs 42, 44, 45 (16297143, 27127523, 24315352), Page 6 Lines 299-302, Ref 76 (18193345)

Added References:

  1. Tamagawa-Mineoka, R., et al., Elevated platelet activation in patients with atopic dermatitis and psoriasis: increased plasma levels of beta-thromboglobulin and platelet factor 4. Allergol Int, 2008. 57(4): p. 391-6.
  2. Bazan-Socha, S., et al., Increased blood levels of cellular fibronectin in asthma: Relation to the asthma severity, inflammation, and prothrombotic blood alterations. Respir Med, 2018. 141: p. 64-71.
  3. Nastałek, M., et al., Plasma platelet activation markers in patients with atopic dermatitis and concomitant allergic diseases. J Dermatol Sci, 2011. 64(1): p. 79-82
  4. Kasperska-Zajac, A. and B. Rogala, Markers of platelet activation in plasma of patients suffering from persistent allergic rhinitis with or without asthma symptoms. Clin Exp Allergy, 2005. 35(11): p. 1462-5.
  5. Koczy-Baron, E., et al., Evaluation of circulating vascular endothelial growth factor and its soluble receptors in patients suffering from persistent allergic rhinitis. Allergy Asthma Clin Immunol, 2016. 12: p. 17.
  6. Potaczek, D.P., Links between allergy and cardiovascular or hemostatic system. Int J Cardiol, 2014. 170(3): p. 278-85.
  7. Kasperska-Zajac, A., Z. Brzoza, and B. Rogala, Platelet activating factor as a mediator and therapeutic approach in bronchial asthma. Inflammation, 2008. 31(2): p. 112-20.

Point 3: What is the chapter “2. Role of Platelets in Allergic Asthma” for? Its title does not differ much from the title of the next section “3. Platelet Contribution in the Pathogenesis of Allergic Asthma”. Should chapter 2 not belong to section 3 as its subsection?

Response 3: We revised the subtitles as follows, i) Current Understanding of Platelets’ Involvement in Allergic Asthma (Page 2 Lines 71-72); ii) Mechanisms of Platelets’ Role in the Pathogenesis of Allergic Asthma (Page 4 Line 179); iii) Antiplatelet Therapies for the Asthma Control (Page 13 Lines 383-384).

Point 4: Two additional comprehensive tables one summarizing human and the other animal most important mechanistic studies on the role of platelets in asthma should be added to make this review much more comprehensive.

Response 4: We added two Tables (Table 1. Various platelet-derived factors in the pathogenesis of allergic asthma in human” in Pages 11-12 and Table 2. Evidence of platelets’ involvement in the pathogenesis of allergic asthma in animal models, Pages 12-13) as suggested.

Additional comments:

Point 1: Please, make sure that absolutely all abbreviations used in figures or tables are explained in their legends.

Response 1: We explained all abbreviations used in figures or tables in the legends.

(see: Figure 1 in Page 8. And Figure 2 in Page 10.)

Round 2

Reviewer 1 Report

The Authors made very substantial efforts to address the comments and to further improve the manuscript, they should be congratulated for that. There remains some editing typos and minor changes. Of note, in AERD the dose of aspirin inducing asthma is not low (but high) - line 223. The figure and tables awill be considerably useful for future readers.

Reviewer 2 Report

All criticism taken and all points answered and changed accordingly.

Reviewer 4 Report

My comments have been addressed well. Thank you.